# Genetic inactivation of zinc transporter SLC39A5 improves liver function and hyperglycemia in obesogenic settings

Shek Man Chim[1], Kristen Howell[1], John Dronzek[1], Weizhen Wu[1], Cristopher Van Hout[1], Manuel AR Ferreira[1], Bin Ye[1], Alexander Li[1], Susannah Brydges[2], Vinayagam Arunachalam[1], Anthony Marcketta[1], Adam E Locke[1], Jonas Bovijn[1], Niek Verweij[1], Tanima De[1], Luca Lotta[1], Lyndon Mitnaul[1], Michelle LeBlanc[1], Regeneron Genetics Center[1], David J Carey[3], Olle Melander[4], Alan Shuldiner[1], Katia Karalis[1], Aris N Economides[1,2]*, Harikiran Nistala[1]*, DiscovEHR collaboration, Regeneron Genetics Center

[1]Regeneron Genetics Center, New York, United States; [2]Regeneron Pharmaceuticals, New York, United States; [3]Geisinger Health System, Danville, United States; [4]Department of Clinical Sciences, Malmö, Sweden

## eLife assessment

This **fundamental** study substantially advances our understanding of the role of zinc in metabolism, specifically a newly established clinical link between mutations in the zinc transporter SLC39A5, elevated serum zinc levels, and a reduction in the risk of Type 2 Diabetes. The provided evidence is **solid** with state-of-the-art genetic analysis of large human cohorts followed by a comprehensive analysis of a mouse SLC39A5 knockout mutant, establishing that SLC39A5 plays a role in hepatic lipid handling through AMPK signaling, although the limited analysis of a pancreatic phenotype that has previously been described constitutes a weakness. This study will be of relevance to researchers interested in metabolism, fatty liver disease, and the biology of trace elements.

*For correspondence:
aris.economides@regeneron.com (ANE);
kiran.nistala@gmail.com (HN)

**Abstract** Recent studies have revealed a role for zinc in insulin secretion and glucose homeostasis. Randomized placebo-controlled zinc supplementation trials have demonstrated improved glycemic traits in patients with type II diabetes (T2D). Moreover, rare loss-of-function variants in the zinc efflux transporter *SLC30A8* reduce T2D risk. Despite this accumulated evidence, a mechanistic understanding of how zinc influences systemic glucose homeostasis and consequently T2D risk remains unclear. To further explore the relationship between zinc and metabolic traits, we searched the exome database of the Regeneron Genetics Center-Geisinger Health System DiscovEHR cohort for genes that regulate zinc levels and associate with changes in metabolic traits. We then explored our main finding using in vitro and in vivo models. We identified rare loss-of-function (LOF) variants (MAF <1%) in *Solute Carrier Family 39, Member 5* (*SLC39A5*) associated with increased circulating zinc ($p=4.9 \times 10^{-4}$). Trans-ancestry meta-analysis across four studies exhibited a nominal association of *SLC39A5* LOF variants with decreased T2D risk. To explore the mechanisms underlying these associations, we generated mice lacking *Slc39a5*. *Slc39a5*[-/-] mice display improved liver function and reduced hyperglycemia when challenged with congenital or diet-induced obesity. These improvements result from elevated hepatic zinc levels and concomitant activation of hepatic AMPK and AKT signaling, in part due to zinc-mediated inhibition of hepatic protein phosphatase activity. Furthermore, under conditions of diet-induced non-alcoholic steatohepatitis (NASH), *Slc39a5*[-/-] mice display significantly attenuated fibrosis and inflammation. Taken together, these results suggest SLC39A5

as a potential therapeutic target for non-alcoholic fatty liver disease (NAFLD) due to metabolic derangements including T2D.

## Introduction

Zinc ($Zn^{2+}$) is an essential trace element with established roles in enzyme biochemistry and other biological processes. Hence, robust homeostatic mechanisms have evolved to maintain physiological levels of zinc and coordinate spatiotemporal demands across various tissues (*Jackson et al., 1982*). Metal transporter proteins encoded by solute carrier (SLC) gene families SLC30 (zinc transporter, ZnT) and SLC39 (Zrt- and Irt-like protein, ZIP) facilitate zinc homeostasis by mediating cellular $Zn^{2+}$ efflux and uptake, respectively (*Dempski, 2012*).

Converging lines of evidence have shown that zinc plays a crucial role in insulin secretion and glucose metabolism. For example, increasing zinc intake improves glycemic control in prediabetics and patients with T2D (*Ranasinghe et al., 2018*). Furthermore, LOF variation in *SLC30A8* (encoding ZnT8, a pancreatic islet zinc transporter) in humans associates with reduced glucose levels and a 65% reduction in T2D risk resulting from enhanced insulin responsiveness to glucose combined with increased pro-insulin processing (*Flannick et al., 2014*; *Dwivedi et al., 2019*). To further explore mechanisms underlying the T2D-protective role of zinc and identify additional genetic determinants influencing systemic zinc homeostasis, we tested loss-of-function variation in zinc transporters for association with circulating zinc and T2D risk and identified rare putative LOF (pLOF) variants (MAF <1%) in *SLC39A5* associated with elevated circulating zinc (p=4.9 × 10⁻⁴). We demonstrate that the identified pLOF variants encode non-functional SLC39A5 proteins. In mice, loss of *Slc39a5* results in elevated hepatic zinc, and lower glucose levels, and has protective effects in models of congenital and diet-induced obesity. These effects appear to be mediated by the activation of hepatic AMPK and AKT signaling, thereby uncovering a mechanistic basis for zinc-induced liver protection and indicating that SLC39A5 inhibition may hold therapeutic potential in NAFLD and T2D (*Figure 1—figure supplement 1*).

## Results

### Rare loss-of-function variants in *SLC39A5* associate with elevated serum zinc and protection from type II diabetes

Using exome sequence data from participants of European ancestry in the Regeneron Genetics Center-Geisinger Health System DiscovEHR study, we identified rare pLOF variants (MAF <1%) in *SLC39A5* associated with increased circulating zinc levels in heterozygous carriers (p=4.9 × 10⁻⁴; *Figure 1A*). We also tested rare LOF variants in *SLC39A5* for association with T2D in a multi-ethnic meta-analysis of four studies (UK Biobank, DiscovEHR, Mount Sinai's BioMe study, and Malmö Diet and Cancer Study), totaling >62,000 cases and >518,000 controls, and found them to be nominally associated with protection from T2D (OR 0.82, 95% CI 0.68–0.99, p=3.7 × 10⁻², *Figure 1B*). Using serum call-back analyses, we confirmed that circulating zinc levels in *SLC39A5* heterozygous loss of function carriers are elevated by 12% as compared to age, sex, and BMI-matched reference controls (p=0.0024; *Figure 1C*). Analyses of insulin production (proinsulin/insulin), insulin clearance (insulin/c-peptide ratio), and blood glucose demonstrated no differences based on genotype (*Figure 1D–I* and *Supplementary file 1*). These results, in conjunction with the lack of *SLC39A5* expression in pancreatic β-cells (*Baron et al., 2016*; *Muraro et al., 2016*), suggest that SLC39A5 does not influence pancreatic β-cell development or function.

To test whether the pLOF variants result in loss of protein function, we first examined their expression and cellular localization by immunofluorescence and flow cytometry. In these analyses we included several observed pLOF variants: p.Y47*(c.141C>G), p.R311*(c.931C>T), and p.R322*(c.964C>T). Bicistronic (IRES-DsRED) mammalian expression constructs encoding untagged wild-type or SLC39A5 muteins (Y47*, R311*, R322*,) were transfected into HEK293 cells (*Figure 1—figure supplement 2A–E*). Consistent with previous reports (*Wang et al., 2004*), flow cytometry and immunofluorescence analyses at steady state demonstrated that wild-type SLC39A5 localized to the cell surface (*Figure 1—figure supplement 2A and B*). In contrast, localization of variants Y47*, R311*, and R322* to the cell surface was reduced by ~91%, 98%, and 99%, respectively (*Figure 1—figure supplement*

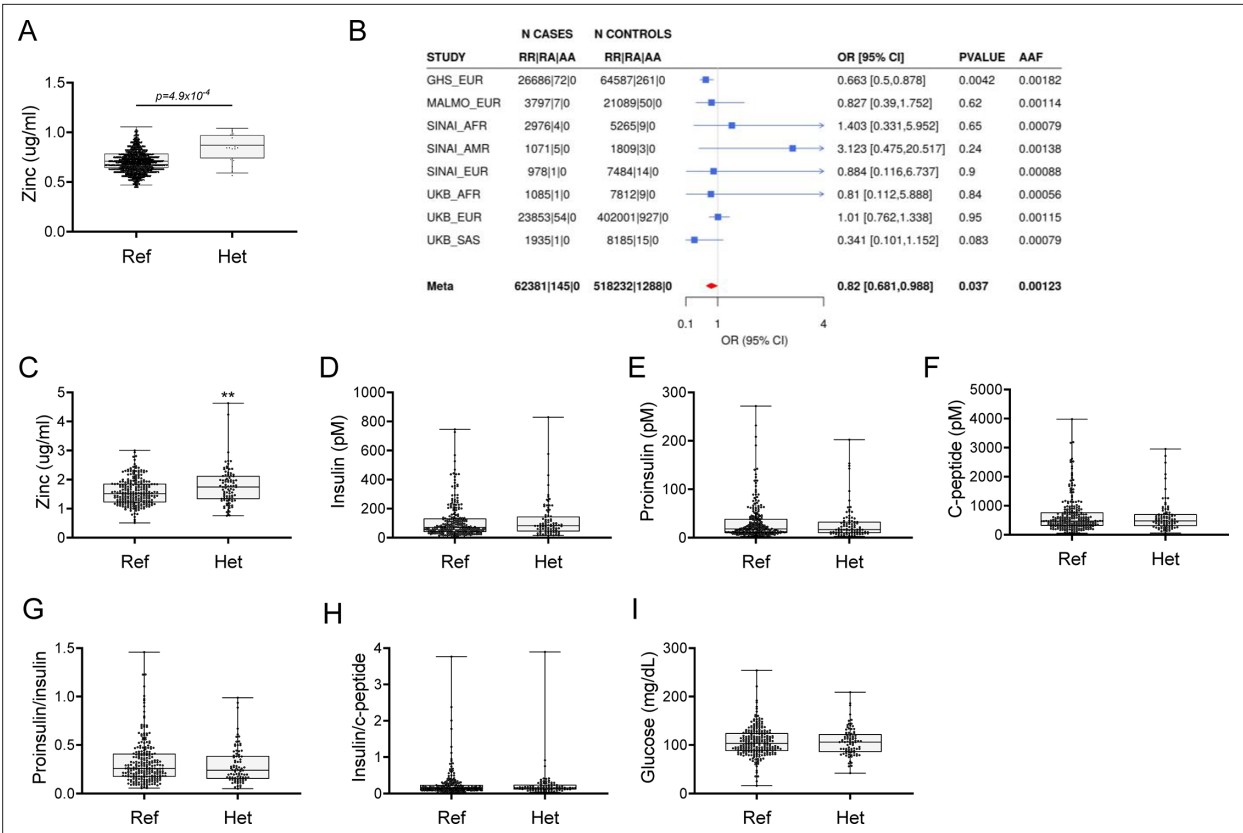

**Figure 1.** Rare putative LOF (pLOF) variants in *SLC39A5* are associated with elevated serum zinc and nominal protection against type II diabetes (T2D). (**A**) Serum zinc in carriers of *SLC39A5* pLOF variants in the discovery cohort. Controls (Ref; *SLC39A5+/+*) and heterozygous carriers of pLOF variant alleles in *SLC39A*5 (Het; *SLC39A5+/-*). Subject numbers: Ref and Het, respectively: n=5317 and n=15. (**B**) Trans-ancestry meta-analysis of the association of *SLC39A5* pLOF variants with T2D. (**C–I**) Serum zinc and insulin profile of age, sex and BMI-matched controls in serum call back study. Subject numbers: Ref and Het, respectively: n=246–253 and n=86–91, **p<0.01, unpaired t-test. Numeric data is summarized in *Supplementary file 1*.

The online version of this article includes the following figure supplement(s) for figure 1:

**Figure supplement 1.** Graphic abstract.

**Figure supplement 2.** *SLC39A5* putative LOF (pLOF) variants p.Y47*(c.141C>G), p.R311*(c.931C>T), and p.R322*(c.964C>T) encode for non-functional proteins.

*2D*). To assess the zinc transport function of these variants, we leveraged a zinc-dependent trans-activation assay using a metal regulatory element (MRE) responsive luciferase reporter. Wild-type SLC39A5 resulted in dose-dependent activation of the reporter to $Zn^{2+}$ (an effect that was attenuated by zinc chelator N,N,N',N'-Tetrakis(2-pyridylmethyl)ethylenediamine) (*Figure 1—figure supplement 2C*), whereas variants Y47*, R311*, R322* failed to mediate a response (*Figure 1—figure supplement 2D and E*). Therefore, variants Y47*, R311*, R322* encode non-functional proteins. Their association with elevated serum zinc levels in the corresponding carriers is consistent with the proposed role of SLC39A5 in maintaining systemic zinc homeostasis by facilitating efflux of excess serosal zinc into the gut lumen (*Wang et al., 2004*).

### *Slc39a5* homozygous-null mice display elevated serum and tissue zinc

To investigate the role of *Slc39a5* in glucose homeostasis in vivo, we generated *Slc39a5*-null mice (*Figure 2—figure supplement 1A*). The resulting *Slc39a5-/-* mice completely lacked *Slc39a5* transcript and protein in their duodenum and liver (*Figure 2—figure supplement 1B and C*) , two tissues with documented expression of SLC39A5 (*Wang et al., 2004*). Consistent with our observations in human heterozygous LOF carriers, *Slc39a5+/-*mice had elevated circulating zinc levels (~26% in females and ~23% in males) compared to wildtype littermates. The elevation in circulating zinc was greatly accentuated in *Slc39a5-/-* mice (~280% in females and ~227% in males) compared to wild-type

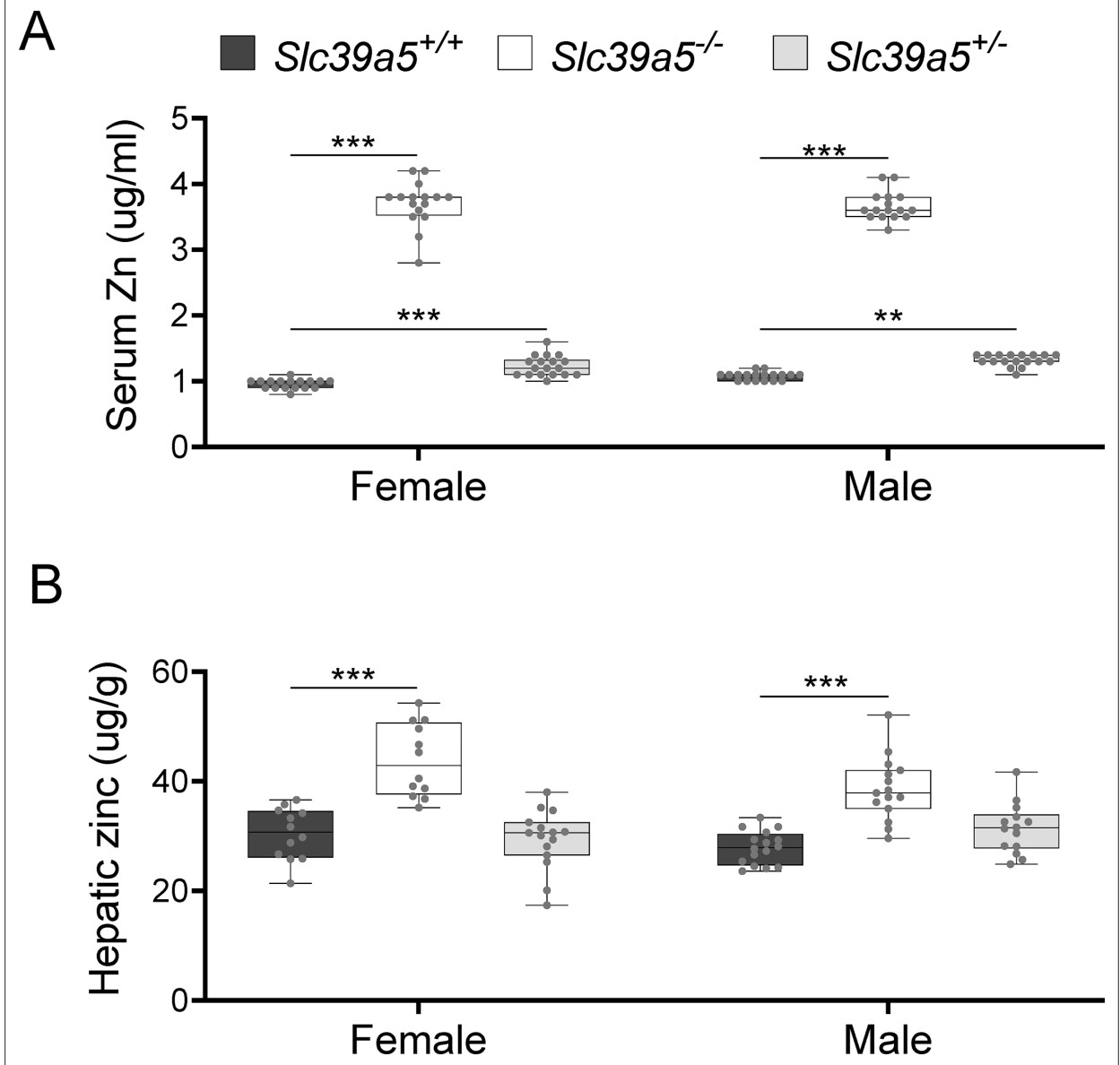

**Figure 2.** Loss of *Slc39a5* results in elevated circulating and hepatic zinc levels in mice. Serum zinc (**A**) and hepatic zinc (**B**) in *Slc39a5+/+*, *Slc39a5-/-*, and *Slc39a5+/-*mice at 40 wk of age, n=16–18. **p<0.01, ***p<0.001, two-way ANOVA with post hoc Tukey's test.

The online version of this article includes the following source data and figure supplement(s) for figure 2:

**Figure supplement 1.** Generation and characterization of the *Slc39a5-/-* mice.

**Figure supplement 1—source data 1.** Original files of the full raw uncropped, unedited blots.

**Figure supplement 1—source data 2.** Figures with the uncropped blots with the relevant bands clearly labelled.

**Figure supplement 2.** Loss of *Slc39a5* does not alter hepatic magnesium, iron, copper, calcium, and cobalt levels in mice challenged with high-fat high fructose diet (HFFD).

littermates (*Figure 2A*). *Slc39a5-/-* mice displayed normal fecundity and had no overt phenotypes even at 22 mo of age. Elemental analyses of major organs (in both sexes) revealed that *Slc39a5-/-* mice had significantly elevated zinc levels in the liver, bone, kidneys, and brain, and lower levels in the pancreas (*Figure 2B* and *Supplementary file 2*). These phenotypes are consistent with previously reported *Slc39a5* knockout mouse models (*Geiser et al., 2013*; *Wang et al., 2019*). No differences in magnesium, iron, copper, cobalt, or calcium were observed in the liver (*Figure 2—figure supplement 2*). Serum chemistry analysis in adult mice (10 mo, both sexes) demonstrated no differences in

pancreatic amylase, renal function parameters (blood urea nitrogen, creatinine, total protein and uric acid), electrolytes (chloride, potassium and sodium), and liver enzymes (alanine aminotransferase; ALT and aspartate aminotransferase; AST) (*Supplementary file 3*), suggesting that the observed changes in tissue zinc levels are physiologically inert at this age. Unlike *Slc39a5*[-/-] mice, *Slc39a5*[+/-] mice showed no changes in tissue zinc levels despite elevation in serum zinc indicating that the free exchangeable pool of serum zinc in the *Slc39a5*[+/-] mice is not sufficient to alter zinc balance within tissues (*Supplementary file 2*). Conservation at the protein level (~83.5% identity), similar postnatal expression (*Dufner-Beattie et al., 2004*), and preserved function between mouse and human orthologs suggest that *Slc39a5*[-/-] mice provide a valid model to explore the observed subthreshold T2D protective effect of *SLC39A5* LOF alleles in humans.

## Loss of *Slc39a5* results in reduced fasting blood glucose in congenital and diet-induced obesity models

To assess whether disruption of Slc39a5 function improves glycemic traits in mice, we challenged the *Slc39a5*[-/-] mice with well-established models of congenital (leptin-receptor deficiency; *Lepr*[-/-] mice) or diet-induced obesity (*Huang et al., 2004*; *King, 2012*). Loss of *Slc39a5* did not alter body weight in either model (*Figure 3A, E, I, M* and *Figure 3—figure supplement 1A*, *Figure 3—figure supplement 2A*, *Figure 3—figure supplement 3A*, *Figure 3—figure supplement 4A*). *Slc39a5*[-/-];*Lepr*[-/-] mice; or *Slc39a5*[-/-] mice on high-fat high fructose diet (HFFD) showed a significant reduction in fasting blood glucose levels as compared to littermate controls (*Figure 3B, F, J, and N* and *Supplementary file 4* and *Supplementary file 5*), but not fasting insulin levels (*Figure 3C, G, K and O*). However, *Slc39a5*[+/-] mice did not show a similar improvement in fasting blood glucose (*Figure 2—figure supplement 1E*), indicating that loss of one copy of *Slc39a5* does not actuate a protective glucose-lowering mechanism; hence, we leveraged *Slc39a5*[-/-] mice for further mechanistic exploration.

Loss of *Slc39a5* in these models demonstrated improved glucose tolerance despite no differences in insulin secretion or clearance at steady state (except female *Slc39a5*[-/-]*; Lepr*[-/-] upon fasting) as compared to littermate controls (*Figure 3—figure supplement 5A–H*, *Figure 3—figure supplement 6A-H*, *Figure 3—figure supplement 1E-F*, *Figure 3—figure supplement 2E-F*, *Figure 3—figure supplement 3E-F*, *Figure 3—figure supplement 4E-F*). Consistently, loss of *Slc39a5* resulted in reduced insulin resistance in these models (*Figure 3D, H, L and P*). Consistent with these observations, no differences in insulin production or clearance were observed in heterozygous carriers of *SLC39A5* LOF variants as compared to age, sex, or BMI-matched reference controls (*Figure 1D–H*) upon serum call-back analyses. Combined with the fact that single-cell transcriptomic data in both humans and mouse show no expression of *SLC39A5* in pancreatic β-cells (*Baron et al., 2016*; *Muraro et al., 2016*; *Almanzar et al., 2020*; *Xin et al., 2016*), these results indicate that the glucose-lowering effects in *Slc39a5*[-/-] mice appear to be independent of pancreatic β-cell function.

## Loss of *Slc39a5* improves liver function

Given that NAFLD and T2D are concurrent comorbidities characterized by hepatic steatosis, glucose intolerance, and insulin resistance (*Tilg et al., 2017*), we explored whether loss of *Slc39a5* and consequent hepatic zinc accumulation (*Figure 2B*) influenced liver function in models of congenital obesity and diet-induced obesity.

*Slc39a5*[-/-];*Lepr*[-/-] mice displayed significant reductions in hepatic lipid accumulation (*Figure 4A* and *Figure 4—figure supplement 1A*), hepatic triglyceride content (*Figure 4B* and *Figure 4—figure supplement 1B*), and in serum ALT and AST levels (biomarkers of liver damage) (*Figure 4C, D*, *Figure 4—figure supplement 1C, D* and *Supplementary file 4*) compared to littermate *Lepr*[-/-] mice. Moreover, *Slc39a5*[-/-]*; Lepr*[-/-] mice displayed reduced NAFLD activity score (an aggregate score of macrovesicular steatosis, hepatocellular hypertrophy, and inflammation) (*Figure 4E* and *Figure 4—figure supplement 1E*). Consistent with reduced lipid burden, expression of hepatic fatty acid synthase expression trended lower in *Slc39a5*[-/-];*Lepr*[-/-] mice (*Figure 3—figure supplement 1B–D*, *Figure 3—figure supplement 2B-D*). Moreover, hepatic and serum beta-hydroxybutyrate levels were elevated in *Slc39a5*[-/-]*; Lepr*[-/-] mice as compared to *Lepr*[-/-] mice, indicative of elevated mitochondrial β-oxidation and disposal of excess hepatic lipid resulting from the leptin receptor deficiency (*Figure 4F*, *Figure 4—figure supplement 1F*, *Figure 3—figure supplement 6K–L*).

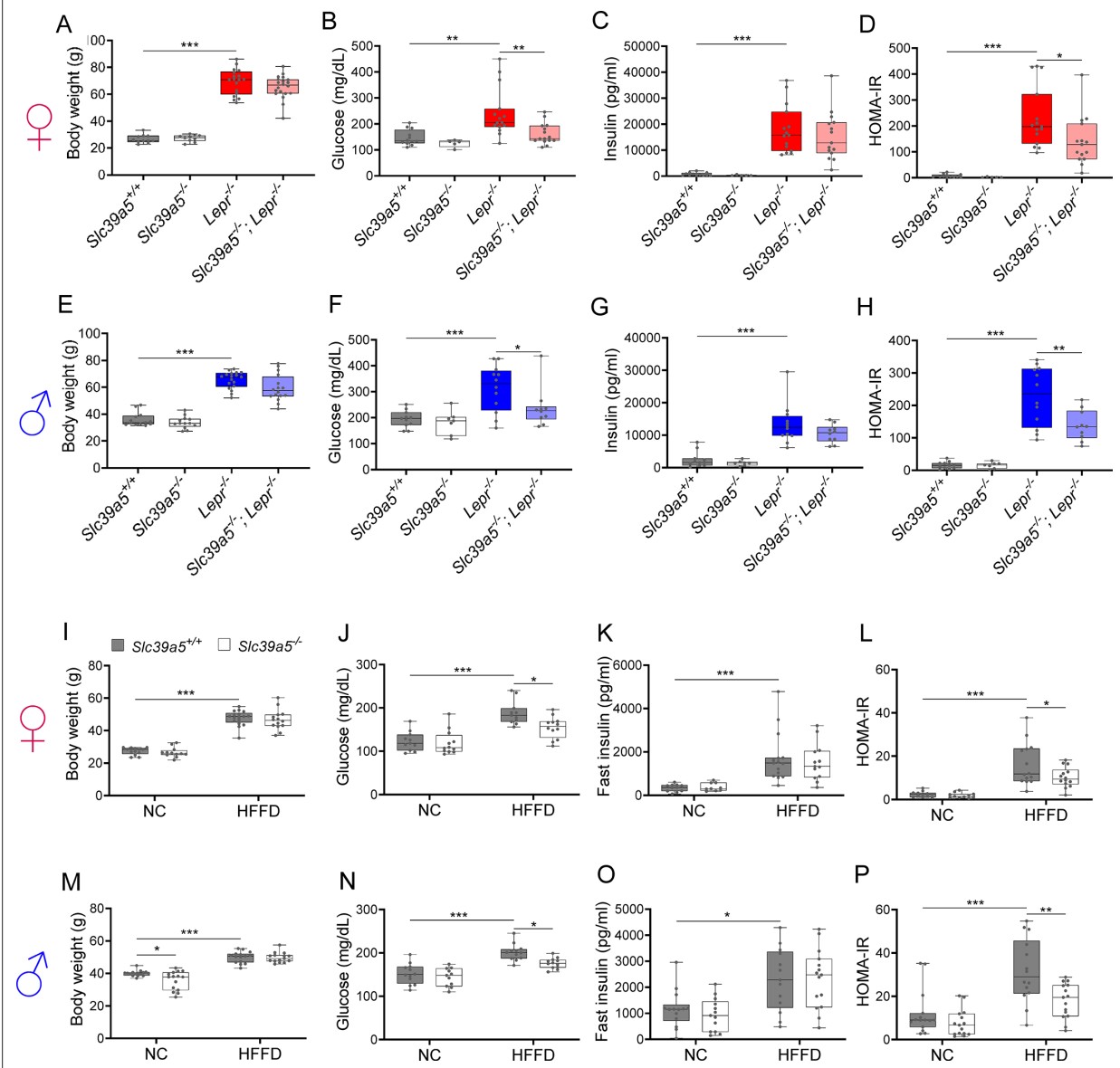

**Figure 3.** Loss of *Slc39a5* improves glycemic traits in leptin-receptor deficient mice and in mice challenged with high-fat high fructose diet (HFFD). Female (A-D, I-L; ♀) and Male (E-H, M-P; ♂) mice. (**A–H**) *Slc39a5-/-;Lepr-/-* and corresponding control mice. (**A, E**) Body weight at 34 wk. (**B, F**) Fasting blood glucose at 34 wk. (**C, G**) Fasting insulin at 34 wk. (**D, H**) Homeostatic model assessment for insulin resistance (HOMA-IR) at 34 wk. *Slc39a5+/+ and Slc39a5-/-* (n=5–12), *Lepr -/-* and *Slc39a5 -/-; Lepr -/-* (n=10–15). *p<0.05, **p<0.01, ***p<0.001, one-way ANOVA with post hoc Tukey's test. (**I–P**) *Slc39a5-/- and Slc39a5+/+* mice were fed HFFD or NC for 30 wk. (**I, M**) Body weight at 30 wk. (**J, N**) Fasting blood glucose at 30 wk. (**K, O**) Fasting insulin at 30 wk. (**L, P**) HOMA-IR at 30 wk, n=11–15. *p<0.05, **p<0.01, ***p<0.001, two-way ANOVA with post hoc Tukey's test. Numeric data is summarized in ***Supplementary file 4*** and ***Supplementary file 5***.

The online version of this article includes the following source data and figure supplement(s) for figure 3:

**Figure supplement 1.** Metabolic profiling of female *Slc39a5-/-; Lepr-/-* mice.

**Figure supplement 1—source data 1.** Original files of the full raw uncropped, unedited blots.

**Figure supplement 1—source data 2.** Figures with the uncropped blots with the relevant bands clearly labelled.

**Figure supplement 2.** Loss of *Slc39a5* reduces hepatic fatty acid synthase expression but does not change the insulin profile of male *Lepr-/-* mice.

**Figure supplement 2—source data 1.** Original files of the full raw uncropped, unedited blots.

**Figure supplement 2—source data 2.** Figures with the uncropped blots with the relevant bands clearly labelled.

**Figure supplement 3.** Loss of *Slc39a5* reduces hepatic fatty acid synthase expression but does not change insulin profile in female mice challenged with high-fat high fructose diet (HFFD).

*Figure 3 continued on next page*

*Figure 3 continued*

**Figure supplement 3—source data 1.** Original files of the full raw uncropped, unedited blots.

**Figure supplement 3—source data 2.** Figures with the uncropped blots with the relevant bands clearly labelled.

**Figure supplement 4.** Loss of *Slc39a5* reduces hepatic fatty acid synthase expression but does not change insulin profile in male mice challenged with high-fat high fructose diet (HFFD).

**Figure supplement 4—source data 1.** Original files of the full raw uncropped, unedited blots.

**Figure supplement 4—source data 2.** Figures with the uncropped blots with the relevant bands clearly labeled.

**Figure supplement 5.** Loss of *Slc39a5* improves glycemic traits in *Lepr-/-* mice and in mice challenged with high-fat high fructose diet (HFFD).

**Figure supplement 6.** Additional data of glucose-stimulated insulin secretion, serum hepatic beta-hydroxybutyrate (BHOB), and pancreas histology in mouse models.

Next, we examined whether loss of *Slc39a5* improves liver function in diet-induced obesity. HFFD significantly increased body weight, serum ALT and AST levels, and NAFLD activity score (***Supplementary file 5***). Loss of *Slc39a5* had no considerable impact on body weight in this model but resulted in marked reductions of hepatic triglyceride content in both sexes (***Figure 4H*** and ***Figure 4—figure supplement 1H***). However, loss of *Slc39a5* resulted in sex-specific differences in most NAFLD-related traits, with females benefiting more significantly compared to males, displaying significant reductions in hepatic steatosis (***Figure 4G***), serum ALT (but not AST) (***Figure 4I and J***), NAFLD activity score (***Figure 4K***), and hepatic fatty acid synthase levels (***Figure 3—figure supplement 3B–D***), and a significant elevation in hepatic and serum beta-hydroxybutyrate levels (***Figure 4L*** and ***Figure 3—figure supplement 6***). Lastly, in contrast to what was observed in $Slc39a5^{-/-};Lepr^{-/-}$ mice, hepatic glucose-6-phosphatase levels were significantly reduced in HFFD female $Slc39a5^{-/-}$ mice (***Figure 3—figure supplement 3B-D***), suggesting that reduced hepatic gluconeogenesis may contribute in part to the observed glucose lowering in these mice.

In HFFD male $Slc39a5^{-/-}$ mice, however, there were no improvements in serum ALT, AST, and NAFLD activity score (***Figure 4—figure supplement 1I–K***), despite reductions in hepatic triglyceride content (***Figure 4—figure supplement 1H***). Significant elevation in hepatic and serum beta-hydroxybutyrate levels and nominal reduction in hepatic fatty acid synthase levels (***Figure 3—figure supplement 4B–D***, ***Figure 3—figure supplement 6J*** and ***Figure 4—figure supplement 1L***) were suggestive of reduced lipid burden in HFFD male $Slc39a5^{-/-}$ mice.

Taken together, these studies suggest that loss of *Slc39a5* in metabolically challenged mice results in reduced hepatic lipid burden and improved hepatic insulin sensitivity, ultimately leading to improved systemic glucose homeostasis.

## Loss of *Slc39a5* results in activation of hepatic AMPK and AKT signaling

To explore the mechanism underlying improved hepatic steatosis and glycemic traits in $Slc39a5^{-/-}$ mice we evaluated two key signaling hubs that mediate lipid metabolism and insulin sensitivity, AMPK and AKT. Activation of hepatic AMPK signaling in a diet-induced obesity model reduces hepatic steatosis and downstream inflammation and fibrosis (***Garcia et al., 2019***), whereas activation of hepatic AKT reduces glucose production in the liver (***Titchenell et al., 2016***). Hence, we evaluated phosphorylation of Thr172 in AMPKα subunit (p.AMPKα) and Ser473 phosphorylation of AKT (p.AKT) in liver lysates from $Slc39a5^{-/-}$; $Lepr^{-/-}$ mice; and HFFD-fed $Slc39a5^{-/-}$ mice and their respective controls. Prior to evaluating p.AMPKα and p.AKT, we confirmed that all $Slc39a5^{-/-}$ mice used in these experiments displayed hepatic zinc accumulation (***Figure 5B and E***, ***Figure 5—figure supplement 1B and E***) and increased expression of the zinc-responsive genes *Mt1* and *Mt2* (***Figure 5C and F***, ***Figure 5—figure supplement 1C and F***, ***Figure 5—figure supplement 2B and E***, ***Figure 2—figure supplement 2B and E***). The p.AMPKα levels were elevated in both $Slc39a5^{-/-}$; $Lepr^{-/-}$ (***Figure 5A***, ***Figure 5—figure supplement 1A*** and ***Figure 5—figure supplement 2A and D***) and HFFD $Slc39a5^{-/-}$ mice (***Figure 5D***, ***Figure 5—figure supplement 1D*** and ***Figure 2—figure supplement 2A and D***) regardless of sex as compared to controls. However, significant hepatic AKT activation was observed only in female $Slc39a5^{-/-}$; $Lepr^{-/-}$; and HFFD $Slc39a5^{-/-}$ mice (***Figure 5A and D***, ***Figure 5—figure supplement 1A and D***, ***Figure 5—figure supplement 2A and D***, ***Figure 2—figure supplement 2A and D***).

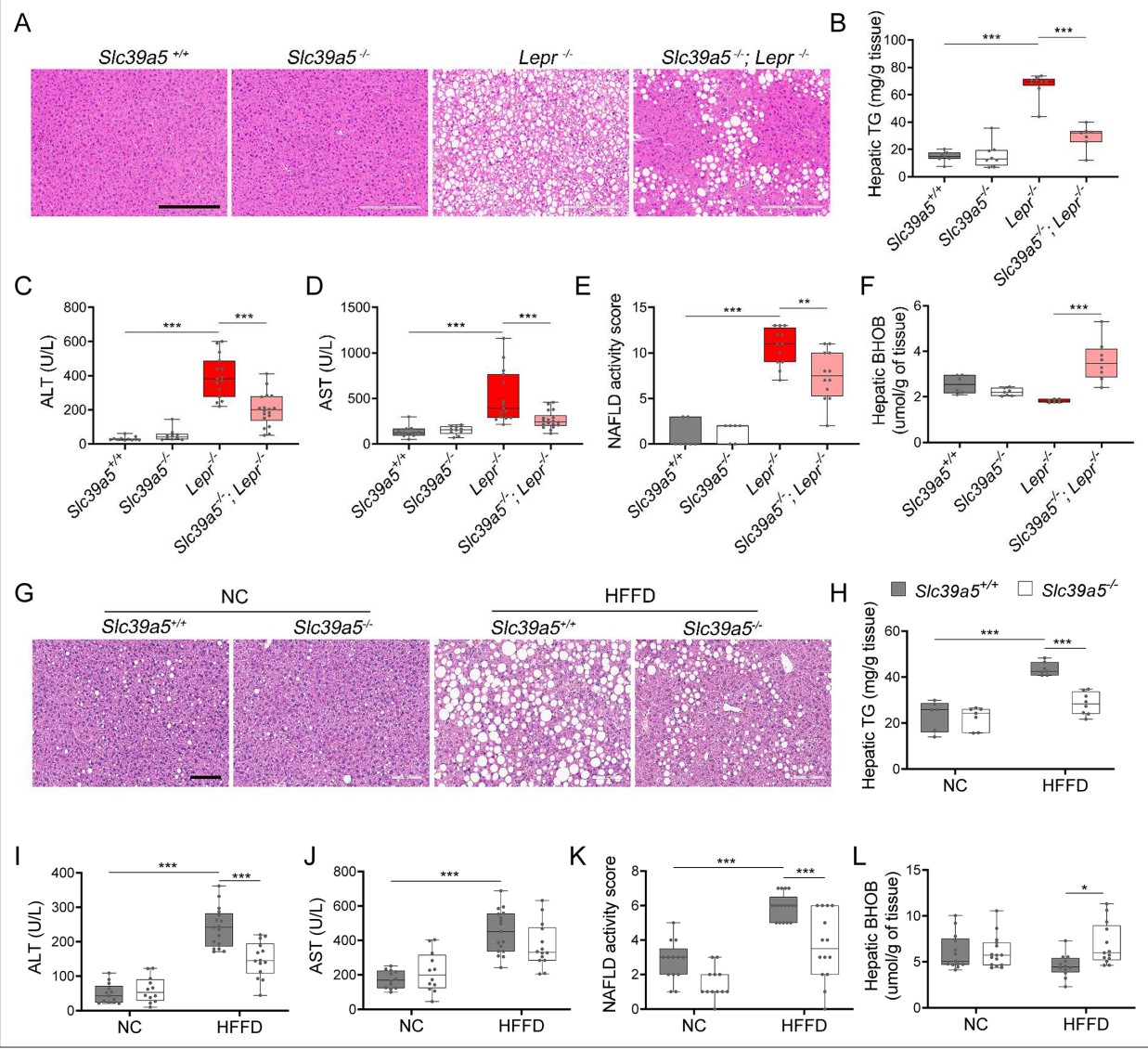

**Figure 4.** Loss of *Slc39a5* improves liver function and steatosis in leptin-receptor deficient female mice and in female mice challenged with high-fat high fructose diet (HFFD). *Slc39a5-/-;Lepr-/-* and corresponding control mice (**A–F**) were sacrificed after 16 hr fasting at 34 wk of age. (**G–L**) *Slc39a5-/-* and *Slc39a5+/+* mice were fed HFFD or NC for 30 wk and sacrificed after 16 hr of fasting. (**A, G**) Representative images of livers stained with H&E. Scale bar, 200 μm. (**B, H**) Hepatic triglyceride (TG) content in explanted liver samples at an endpoint. (**C, I**) Serum ALT. (**D, J**) Serum AST. (**E, K**) Non-alcoholic fatty liver disease (NAFLD) activity score, (**F, L**) Hepatic beta-hydroxybutyrate (BHOB). *p<0.05, **p<0.01, ***p<0.001, *Slc39a5-/-;Lepr-/-* and corresponding control mice: one-way ANOVA with post hoc Tukey's test, HFFD or NC: two-way ANOVA with post hoc Tukey's test. Numeric data is summarized in *Supplementary file 4* and *Supplementary file 5*.

The online version of this article includes the following figure supplement(s) for figure 4:

**Figure supplement 1.** Loss of *Slc39a5* improves liver function and steatosis in *Lepr-/-* male mice and reduces hepatic triglyceride in male mice challenged with high-fat high fructose diet (HFFD).

To further explore the potential role of elevated hepatic zinc in AMPK and AKT activation, we examined whether exogenous zinc activates AMPK and AKT signaling in primary human hepatocytes. Zinc activated AKT signaling in these cells in a dose-dependent manner with no adverse effect on cell viability, whereas magnesium had no effect (*Figure 5G and H*, *Figure 5—figure supplement 3A*). Moreover, zinc-activated AMPK signaling and its downstream substrates acetyl-CoA carboxylase (ACC), and liver kinase B1 (LKB1; the kinase responsible for AMPKα Thr172 phosphorylation) (*Figure 5G and H*). Time-resolved analyses of zinc-mediated activation of LKB, AMPK, and AKT indicated that zinc activates AMPK and AKT signaling acutely (within 4 hr) suggesting that zinc influences

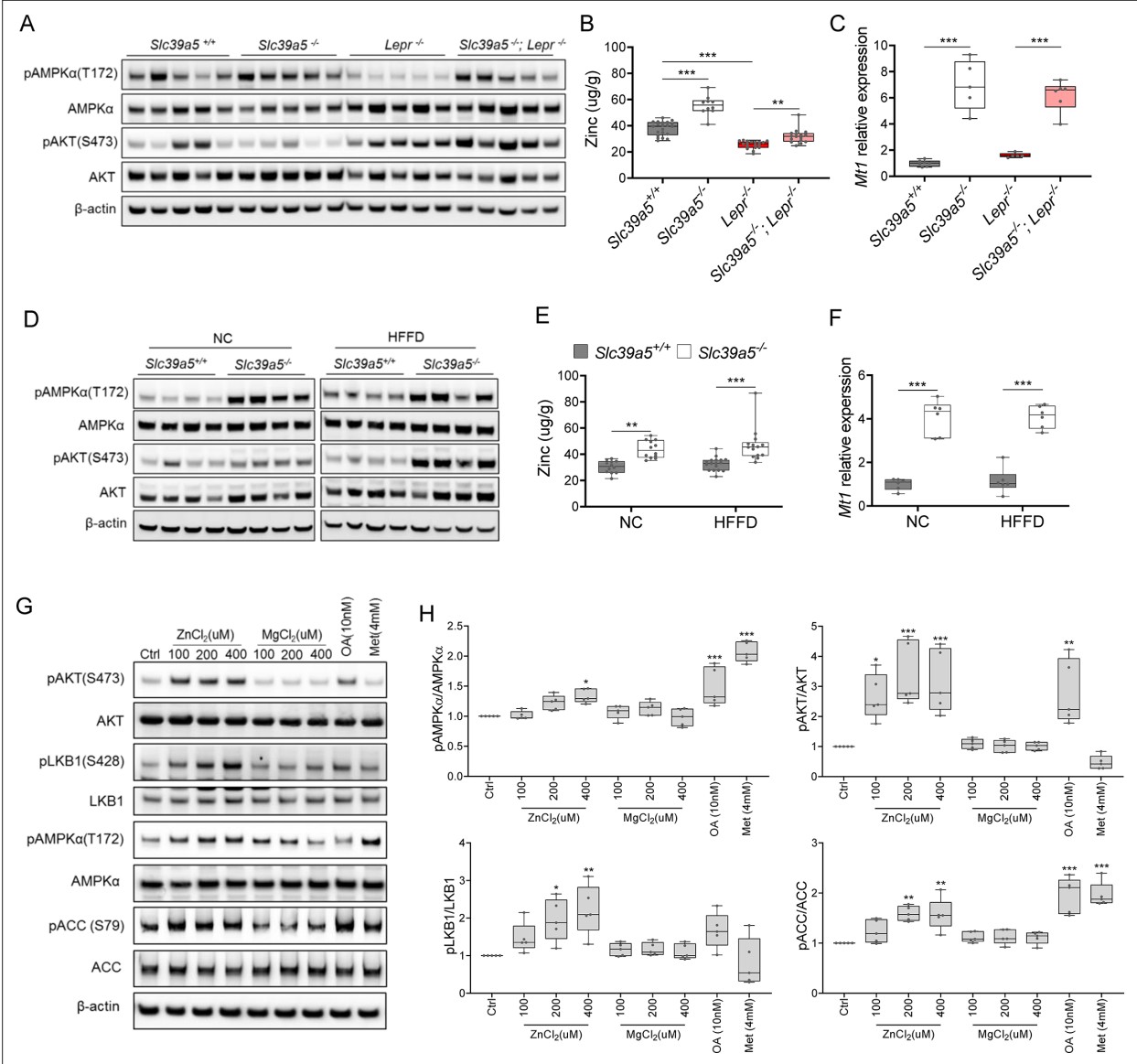

**Figure 5.** Loss of *Slc39a5* results in elevated hepatic zinc and activation of hepatic AMPK signaling in leptin-receptor deficient female mice and female mice challenged with high-fat high fructose diet (HFFD). Analyses were done on explanted liver samples collected after 16 hr of fasting at an endpoint in *Lepr*[-/-] (**A–C**) and HFFD mice (**D–F**). (**A, D**) Immunoblot analysis of hepatic AMPK and AKT activation. AMPK and AKT signaling is activated in *Lepr*[-/-]; *Slc39a5*[-/-] mice and HFFD *Slc39a5*[-/-] mice (compared to their *Scl39a5*[+/+] counterparts). (**B, E**) Hepatic zinc is elevated in *Lepr*[-/-]; *Slc39a5*[-/-] mice and HFFD *Slc39a5*[-/-] mice (n=10–21). (**C, F**) Elevated hepatic zinc results in increased *Mt1* (zinc responsive gene) expression in both models. (**G**) Immunoblot analysis of primary human hepatocytes treated with zinc chloride ($ZnCl_2$), and magnesium chloride ($MgCl_2$), okadaic acid (OA), metformin (Met) for 4 hr. Zinc-activated AMPK and AKT signaling in primary human hepatocytes. (**H**) Densitometric analysis of immunoblots (compared to control). *$p<0.05$, **$p<0.01$, ***$p<0.001$, ANOVA with post hoc Tukey's test.

The online version of this article includes the following source data and figure supplement(s) for figure 5:

**Source data 1.** Original files of the full raw uncropped, unedited blots.

**Source data 2.** Figures with the uncropped blots with the relevant bands clearly labeled.

**Figure supplement 1.** Loss of *Slc39a5* results in elevated hepatic zinc and activation of hepatic AMPK signaling in congenital and diet-induced obesity models.

**Figure supplement 1—source data 1.** Original files of the full raw uncropped, unedited blots.

**Figure supplement 1—source data 2.** Figures with the uncropped blots with the relevant bands clearly labeled.

**Figure supplement 2.** Loss of *Slc39a5* does not alter hepatic magnesium, iron, copper, calcium, and cobalt levels in *Lepr-/-* mice.

**Figure supplement 3.** Zinc activates AMPK and AKT signaling in a time-dependent and dose-dependent manner.

*Figure 5 continued on next page*

*Figure 5 continued*

**Figure supplement 3—source data 1.** Original files of the full raw uncropped, unedited blots.

**Figure supplement 3—source data 2.** Figures with the uncropped blots with the relevant bands clearly labeled.

**Figure supplement 4.** Elevated hepatic zinc results in reduced protein phosphatase activity.

phosphorylation of these substrates independent of de novo protein synthesis (*Figure 5—figure supplement 3B*). Similar results were obtained in the human hepatoma cell line HepG2 (*Figure 5—figure supplement 3C and D*).

Zinc is a potent inhibitor of protein phosphatases, including protein phosphatase 2 A (PP2A) and protein tyrosine phosphatase-1B (PTP1B) (*Bellomo et al., 2016*; *Xiong et al., 2013*), both of which regulate the phosphorylation of AMPKα. Liver-specific ablation of *Ppp2ca* (encoding PP2A's catalytic subunit) improves glucose tolerance and insulin sensitivity in mice (*Xian et al., 2015*), whereas liver-specific ablation of *Ptpn1* (encoding PTP1B) improves glucose tolerance, insulin sensitivity, and lipid metabolism (*Delibegovic et al., 2009*). Given that hepatic zinc is elevated in *Slc39a5$^{-/-}$* mice, we evaluated hepatic phosphoserine/threonine (p.Ser/Thr) and phosphotyrosine (p.Tyr) phosphatase activity in the congenital and diet-induced obesity mice at the endpoint. *Slc39a5$^{-/-}$;Lepr$^{-/-}$* mice displayed reduced p.Ser/Thr and p.Tyr phosphatase activity compared to *Lepr$^{-/-}$* littermate controls (*Figure 5—figure supplement 4A and B*). Under HFFD, female *Slc39a5$^{-/-}$* mice showed reduced hepatic p.Ser/Thr and p.Tyr phosphatase activity (33% and 28%, respectively), and non-statistically significant reductions were also observed in male *Slc39a5$^{-/-}$* mice (*Figure 5—figure supplement 4C and D*). Consistent

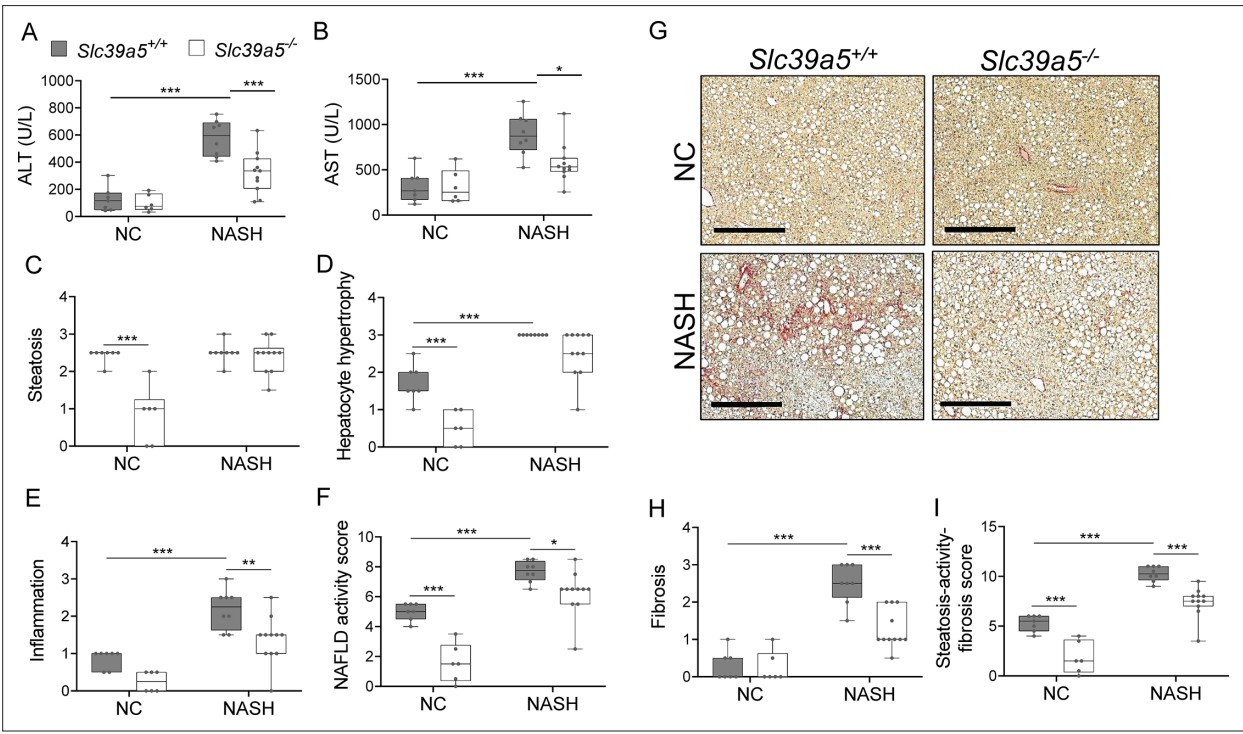

**Figure 6.** Loss of *Slc39a5* improves hepatic inflammation and fibrosis in female mice challenged with diet-induced non-alcoholic steatohepatitis (NASH). *Slc39a5-/- and Slc39a5+/+* mice were placed on a NASH-inducing diet or NC for 40 wk and sacrificed after 16 hr of fasting. (**A, B**) NASH *Slc39a5$^{-/-}$* mice display reduced serum ALT and AST levels. (**C–E**) Histology scores for steatosis, hepatocyte hypertrophy, and inflammation. (**F**) NAFLD activity score was reduced in NASH *Slc39a5-/-* mice. (**G–I**) NASH *Slc39a5$^{-/-}$* mice display reduced fibrosis. (**G**) Representative images of explanted livers sample stained with picrosirius red indicative of collagen deposition. Scale bar, 300 µm. (**H, I**) Fibrosis and steatosis-activity-fibrosis scores. n=6–7 (NC) and 8–11 (NASH), *p<0.05, **p<0.01, ***p<0.001, two-way ANOVA with post hoc Tukey's test. Numeric data is summarized in *Supplementary file 6*.

The online version of this article includes the following figure supplement(s) for figure 6:

**Figure supplement 1.** Loss of *Slc39a5* reduces hepatic inflammation and fibrosis in male mice challenged with diet-induced non-alcoholic steatohepatitis (NASH).

**Figure supplement 2.** Loss of *Slc39a5* improves liver function in mice challenged with diet-induced non-alcoholic steatohepatitis (NASH).

with these observations, exogenous zinc inhibited p.Ser/Thr and p.Tyr phosphatase activity in primary human hepatocytes in a dose-dependent manner (*Figure 5—figure supplement 4E*). These results point to zinc-mediated inhibition of protein phosphatase activity as a likely mechanism underlying hepatic AMPK and AKT activation in *Slc39a5*$^{-/-}$ mice.

## Loss of *Slc39a5* reduces hepatic inflammation and fibrosis upon a NASH dietary challenge

NAFLD encompasses a continuum of liver conditions from non-alcoholic fatty liver characterized by steatosis, to non-alcoholic steatohepatitis (NASH) characterized by inflammation and fibrosis (*Friedman et al., 2018*). The improvements in liver function and steatosis in congenital and diet-induced obesity mouse models lacking *Slc39a5*, led us to investigate whether loss of *Slc39a5* protects against NASH. Diet-induced NASH significantly increased serum ALT and AST levels (*Figure 6A and B*, *Figure 6— figure supplement 1A and B*), body weight, fasting blood glucose (*Figure 6—figure supplement 2A, B, F, G*), and liver fibrosis (*Figure 6H and I*, *Figure 6—figure supplement 1H and I*) in *Slc39a5*$^{+/+}$ mice (*Supplementary file 6*). In contrast, *Slc39a5*$^{-/-}$ mice challenged with diet-induced NASH displayed significant reductions in serum ALT and AST levels (*Figure 6A and B*, *Figure 6—figure supplement 1A and B*) and fasting blood glucose (*Figure 6—figure supplement 2B and G*), along with significant improvements in hepatic inflammation and fibrosis (*Figure 6E and H*, *Figure 6—figure supplement 1E and H*) and the expected increases in serum and hepatic zinc (*Figure 6—figure supplement 2C, D, H, I*). Consistently, hepatic collagen deposition (*Figure 6G*, *Figure 6—figure supplement 1G*) were significantly reduced in NASH *Slc39a5*$^{-/-}$ mice. However, NASH *Slc39a5*$^{-/-}$ mice were not protected from hepatic steatosis or hepatocyte hypertrophy (*Figure 6C and D*, *Figure 6—figure supplement 1C and D*). NAFLD activity score and steatosis-activity-fibrosis score (sum of NAFLD activity score and fibrosis score) were significantly reduced in female NASH *Slc39a5*$^{-/-}$ mice, but not in their male counterparts (*Figure 6F and I* and *Figure 6—figure supplement 1F and I*). Nonetheless, hepatic superoxide dismutase (SOD) activity was significantly elevated in both sexes in NASH *Slc39a5*$^{-/-}$ mice (*Figure 6—figure supplement 2E and J*), suggesting that the increase in hepatic zinc may be ameliorating the increased hepatic oxidative stress observed in NASH (*Friedman et al., 2018*).

In aggregate, these studies suggest that the favorable metabolic profile in the *Slc39a5*$^{-/-}$ mice results from convergent hepatoprotective effects due to reduced lipotoxic and oxidative stress.

## Discussion

Zinc is a required trace element for many biological processes. Hence, homeostatic mechanisms have evolved to maintain optimal zinc levels across tissues (*Jackson et al., 1982*). This regulation is accomplished by multiple transporters encoded by the *SLC30* and *SLC39* gene families (*Dempski, 2012*). Given the apparent complexity of the system, we chose to take a human genetics approach to search for zinc transporter genes associated with metabolic traits and discovered a novel association of LOF variants in *SLC39A5* with increased circulating zinc (p=$4.9 \times 10^{-4}$) and a reduced risk of T2D (OR 0.82, 95% CI 0.68–0.99, p=$3.7 \times 10^{-2}$).

To firm up this association and explore underlying molecular mechanisms, we generated mice lacking *Slc39a5*. In line with the human data, and consistent with the proposed role of *Slc39a5* as a non-redundant cell surface zinc transporter facilitating endogenous zinc excretion (*Wang et al., 2004*; *Geiser et al., 2013*; *Dufner-Beattie et al., 2004*), there was significant zinc accumulation in *Slc39a5*$^{-/-}$ mice across several tissues including liver (*Supplementary file 2*). However, there was no significant accumulation of zinc in tissues of *Slc39a5* heterozygous-null mice on a zinc-adequate diet despite significant increases in serum zinc (~26% in females and ~23% in males; *Figure 2*), suggesting that the relative increase of serum zinc in heterozygotes (albeit significant) is insufficient to increase zinc levels in tissues with substantial zinc stores such as the liver.

Nonetheless, given the connection with protective effects arising from heterozygous loss of *SLC39A5* in humans, we examined the effect of loss of *Slc39a5* in mice under conditions of metabolic stress, employing models of congenital or diet-induced obesity, and NASH. We demonstrate that in all three models, loss of *Slc39a5* (in homozygosis) has protective effects that arise from elevation of circulating and hepatic zinc levels. In congenital or HFFD-induced obesity, there was improvement in glycemic traits and liver function, and a reduction of steatosis, which were not accompanied by

reductions in body weight or changes in insulin profile. In a model of diet-induced NASH, loss of *Slc39a5* reduced hepatic inflammation and fibrosis, but without significant changes in steatosis. Mechanistically, these protective effects result at least in part from the inhibition of protein phosphatases (as a result of elevated levels of zinc), and the consequent increase in hepatic AMPK and AKT activation.

The observed protective metabolic effects appear to be extra-pancreatic in both mice and humans, as supported by several lines of evidence. Carriers of heterozygous LOF mutations in *SLC39A5* have elevated serum zinc but exhibit no differences in insulin production or clearance as compared to age, sex, and BMI-matched homozygous reference controls (*Figure 1C–E*). As in humans, loss of *Slc39a5* in mice results in elevated serum zinc (*Figure 2A*) without impairment in pancreatic function (*Supplementary file 3*). Moreover, the observed antihyperglycemic effects in *Slc39a5$^{-/-}$* mice are not driven by changes in insulin production or clearance (*Figure 3—figure supplement 1*, *Figure 3—figure supplement 2*, *Figure 3—figure supplement 3*, *Figure 3—figure supplement 4*). Taken together, these observations suggest that the protective metabolic changes are extra-pancreatic.

Furthermore, our data strongly indicates that the protective effects of loss of *Slc39a5* are actuated by elevated hepatic zinc concentrations. Several lines of evidence support this interpretation. First, metabolic challenges in the form of congenital or diet-induced obesity in mice revealed hepatic zinc deficiency (*Figure 5B*, *Figure 5—figure supplement 1B*) along with associated comorbidities including hepatic steatosis, increased fasting blood glucose, and impaired insulin sensitivity (*Figure 3*). Second, loss of *Slc39a5* in these models resulted in the accumulation of serum zinc and hepatic zinc and concomitant improvement in liver function (*Figure 4*, *Figure 5*, *Figure 4—figure supplement 1*) and systemic glucose homeostasis (*Figure 3*, *Figure 3—figure supplement 5*). These data are consistent with observations that zinc deficiency is associated with obesity (*Marreiro et al., 2002*) and is a biochemical hallmark of fatty liver disease in both rodents and humans *Mohammad et al., 2012*; conversely, zinc supplementation reverses manifestations of zinc deficiency in fatty liver disease and long-term oral zinc supplementation can support liver function and prevent hepatocellular carcinoma development in patients with chronic liver diseases (*Hosui et al., 2018*).

The importance of hepatic zinc in the protective effects against obesity and NASH is further supported by our findings that elevated hepatic zinc in *Slc39a5$^{-/-}$* mice enhanced hepatic AMPK and hepatic AKT signaling (*Figure 5* and *Figure 5—figure supplement 1*). These increases correlated with reductions in hepatic p.Ser/Thr phosphatase and p.Tyr phosphatase levels in both diet-induced and congenital obesity models (*Figure 5—figure supplement 4A–D*), and were corroborated by in vitro evidence (*Figure 5—figure supplement 4E–F*). These findings mirror prior studies showing that zinc inhibits protein serine/threonine and tyrosine phosphatases that dephosphorylate AMPK and AKT (*Bellomo et al., 2016*; *Bellomo et al., 2018*; *Lee et al., 2009*; *Liangpunsakul et al., 2010*; *Galic et al., 2005*; *Krishnan et al., 2018*). In turn, in states of lipotoxic stress, serine/threonine phosphatases such as PP2A and PP2C inhibit AMPK resulting in a feed-forward effect of the lipid overload (*Chen et al., 2017*; *Wang and Unger, 2005*), whereas protein tyrosine phosphatases including PTP1B, TCPTP, and PTEN have been implicated in systemic glucose homeostasis by regulating the PI3K-AKT pathway (*Delibegovic et al., 2009*; *Galic et al., 2005*; *Pal et al., 2012*; *Tsou and Bence, 2012*).

Overall, our studies indicate that the favorable metabolic profile observed in the *Slc39a5$^{-/-}$* mice results from the loss of endogenous zinc excretion and concomitant systemic zinc redistribution. Our study provides for the first-time genetic evidence demonstrating the protective role of zinc against hyperglycemia and unravels the mechanistic basis underlying this effect. Taken together, these observations suggest SLC39A5 inhibition as a potential therapeutic avenue for T2D, and other indications where zinc supplementation alone is inadequate.

# Materials and methods
## Human genetic studies and phenotyping

The Geisinger Health System DiscovEHR study is a hospital-based cohort of patients of the GHS, a large healthcare provision network in Central and Eastern Pennsylvania, United States. More than 200,000 health system participants have been enrolled and >145,000 have had exome sequencing performed by Regeneron Genetics Center. Type 2 diabetes (T2D) cases in DiscovEHR were defined as individuals with an ICD9 (code 250) or ICD10 (code E11) code for T2D, and either a median HbA1c value greater than or equal to 6.5%, or with a prescription for any diabetic medication. Individuals

were excluded from the case pool if they had both an ICD10 code for type 1 diabetes (T1D; code E10) and if they did not have a prescription for any oral hypoglycemic medication. Controls were defined as individuals with no ICD codes for T1D or T2D, a median HbA1c value of less than 5.7%, and with no record of a prescription for any diabetic medication.

The UK Biobank is a prospective biomedical study of ~500,00 adults from across the UK, including extensive phenotype measures and genomic data. T2D in the UK Biobank was defined in line with a previously reported definition in this cohort (*Eastwood et al., 2016*; *Lotta et al., 2018*). The UKB self-reported data were used to identify individuals with 'probable type 2 diabetes,' 'possible type 2 diabetes,' 'probable type 1 diabetes' or 'possible type 1 diabetes,' using a previously published algorithm (*Eastwood et al., 2016*). T2D cases were defined as individuals with 'probable type 2 diabetes' on self-report, or an ICD10 code E11 for T2D. Individuals were excluded from the analysis if they had 'probable type 1 diabetes,' 'possible type 1 diabetes,' or ICD10 code E10, for T1D.

The BioMe study (SINAI) is a highly diverse electronic health record (EHR)-linked biobank of over 50,000 participants from the Mount Sinai Health System (MSHS) in New York, NY. T2D cases in BioMe were defined as individuals meeting at least two of the following three criteria: (1) ICD10 code for T2D (code E11 and/or O24.1), (2) a blood value in keeping with diabetes (median HbA1c value greater than or equal to 6.5% and/or median random glucose greater than or equal to 200 mg/dL), and (3) a prescription for any diabetic medication. Individuals were excluded from the case pool if they had an ICD10 code for T1D (code E10 and/or O24.0) or if they had a record of having received an outpatient prescription for insulin (and no record of other antidiabetic medication). Controls were defined as individuals with no ICD10 codes for any type of diabetes mellitus or a family history of diabetes, median HbA1c value of less than 5.7%, median random glucose of less than 200 mg/dL, no oral glucose tolerance test in pregnancy exceeding a diagnostic threshold for gestational diabetes, and no record of a prescription for any diabetic medication.

The Malmö Diet and Cancer Study (MDCS) is a prospective study of ~53,000 adults living in Malmö, Sweden (*Berglund, 1993*). T2D cases in MDCS were defined as individuals meeting at least two of the following four criteria: (1) ICD10 code for T2D (code E11 and/or O24.1) or T2D noted in diabetes registries, (2) a blood value in keeping with diabetes (HbA1c value greater than or equal to 6.5% and/or fasting glucose greater than or equal to 126 mg/dL), (3) a prescription for non-insulin diabetic medication, and (4) a record of a non-specific diabetes event (e.g. reported at baseline, or extracted from a registry) with an age at diagnosis, or the start of treatment, of greater than or equal to 35 y. Individuals were excluded from the case pool if they had an ICD10 code for T1D (code E10 and/or O24.0), or T1D noted in a diabetes registry, or if they had a record of having received insulin with no record of other antidiabetic medication. Controls were defined as individuals with no ICD10 codes for any type of diabetes mellitus, no family history of diabetes, no other variables indicating a potential diagnosis of diabetes, HbA1c value of less than 5.7%, fasting glucose of less than 100 mg/dL, and no record of a prescription for any diabetic medication.

## Association analyses

Rank-based inverse normal transformed (RINT) quantitative measures (including all subjects and sex-stratified models) with non-missing phenotype information were assessed using an additive mixed model implemented in REGENIE v2 (*Mbatchou et al., 2021*). Prior to normalization, traits were adjusted for a standard set of covariates including age, $age^2$, sex, age ×sex, $age^2$ ×sex, 10 common variant genetic principal components, and 20 genetic principal components derived from rare variants. Binary outcomes were similarly adjusted for age, $age^2$, sex, age ×sex, $age^2$ ×sex, 10 common variant genetic principal components, and 20 genetic principal components derived from rare variants and tested for association using a generalized mixed model implemented in REGENIE v2. Following analysis within each cohort, we performed inverse variance-weighted meta-analysis for T2D using METAL.

## GHS serum call back study

As an orthogonal biochemical assessment of the EHS-reported blood analyte data, a serum call-back study was designed to evaluate serum zinc, blood glucose, insulin synthesis (proinsulin/insulin ratio), and clearance (insulin/c-peptide ratio) in heterozygous carriers of SLC39A5 pLOF variants. Carriers of pLOF variants in SLC39A5 among exome-sequenced participants of European ancestry in

the Regeneron Genetics Center-Geisinger Health System DiscovEHR study were included. Controls included non-carriers of pLOF variants in SLC39A5 and SLC30A8 or the common T2D risk variant rs13266634 in SLC30A8 and non-carrier first-degree relatives of study subjects. Participants with T1D or T2D diagnoses were excluded. Furthermore, two non-carriers were selected for each carrier matching sex, age (+/-5 y), and BMI (+/-5). A total of 22 SLC39A5 LOF variants in 131 carriers and 262 matched non-carriers were identified, however sample (frozen fasting serum) availability limited analyses to ~250 non-carriers and ~90 carriers as shown in *Supplementary file 1*. Serum insulin was measured using a Human Insulin ELISA kit (Millipore, EZHI-14BK), proinsulin using a Human Total Proinsulin ELISA kit (Millipore, EZHPI-15BK), and c-peptide using a Human c-peptide ELISA kit (Abcam, ab178611). Serum zinc was measured using flame atomic absorption spectroscopy as described below. Blood glucose was evaluated using ADVIA Chemistry Glucose Hexokinase_3 reagents (REF 050011429) on a Siemens ADVIA Chemistry XPT analyzer.

## Generation of *Slc39a5* loss of function mice

The genetically engineered *Slc39a5*$^{-/-}$ mouse strain was created using Regeneron's VelociGene technology (*Poueymirou et al., 2007*; *Valenzuela et al., 2003*). Briefly, C57Bl/6NTac embryonic stem cells (ESC) were targeted for ablation of a portion of *Slc39a5*, beginning just after the initiating ATG and ending 5 base pairs before the 3' end of coding exon 2. This region contains the SLC39A5 signal peptide and much of the N-terminal extracellular domain. A lacZ reporter module was inserted in the frame with *Slc39a5*'s initiating Methionine codon, followed by a self-deleting fLoxed neomycin resistance (neo) cassette for selection in mouse C57BL/6NTac embryonic stem cells. The targeted cells were microinjected into 8 cell embryos from Charles River Laboratories Swiss Webster albino mice, yielding F0 VelociMice that were 100% derived from the targeted cells (*Poueymirou et al., 2007*). These mice were subsequently bred to F1, at which point the self-deleting neo cassette was also removed in the male germline. F1 heterozygotes were utilized to generate experimental cohorts, including Slc39a5$^{-/+}$ heterozygous mice and wild-type littermates that were used as controls; this line was maintained in Regeneron's animal facility in the C57Bl/6NTac genetic background throughout the study.

## Animal studies

Mice homozygous for *Slc39a5* loss of function and wild-type littermates were co-housed in a controlled environment (12 hr light/dark cycle, 22 ± 1°C, 60–70% humidity) and fed ad-libitum. All studies were performed in both sexes. For the HFFD study, ten-week-old mice were fed the HFFD diet (46 kcal% Fat, 30kcal% Fructose, TestDiet 5WK9) or a control diet (TestDiet 58Y2) for 30 wk. For the NASH study, ten-week-old mice were fed the NASH diet (40 kcal% Fat, 20 kcal% Fructose, and 2% Cholesterol, ResearchDiets D09100310) or a control diet (ResearchDiets D09100304) for 40 wk. Both HFFD and NASH diets contain ~34 ppm zinc as described in diet spec sheets, and further confirmed by flame atomic absorption spectrometry. *Slc39a5*$^{-/-}$; *Lepr*$^{-/-}$ mice and corresponding control mice (*Slc39a5*$^{+/+}$; *Lepr*$^{-/-}$, *Slc39a5*$^{-/-}$; *Lepr*$^{+/+}$, and *Slc39a5*$^{+/+}$; *Lepr*$^{+/+}$ mice) were fed a normal chow (LabDiet 5053, containing 87 ppm zinc) for 34 wk. All mice used in this study were housed in a pathogen-free environment at Regeneron Pharmaceuticals Inc animal research facility. Sterile water and show were given ad libitum.

## Serum analysis

Sera were collected upon an overnight fast (16 hr). The liver and lipid profile were analyzed using Siemens ADVIA Chemistry XPT analyzer which is maintained and operated according to Siemens' guidelines. The liver and lipid profile contains the following reagents: Alanine Aminotransferase (ALT, Siemens REF 03036926), Aspartate Aminotransferase (AST, Siemens REF 07499718), Cholesterol (CHOL, Siemens REF 10376501), Direct HDL Cholesterol (DHDL, Siemens REF 07511947), LDL Cholesterol Direct (DLDL, Siemens REF 09793248), Non-Esterified Fatty Acids (NEFA, Wako 999–34691, 995–34791, 991–34891, 993–35191), Triglycerides TRIG, Siemens REF 10335892. When mixed with the sample, reagents undergo a colorimetric change proportional to the concentration of the specific analyte. The absorbance is then measured with a halogen light source and used to determine concentration. Serum was also collected for ELISA analysis of proinsulin (Mercodia, #10-1232-01) per manufacturer's guidelines. Briefly, samples were incubated with enzyme conjugate at room temperature for

2 hr and washed. Substrate TMB was added and the reaction was allowed to proceed for 30 min at room temperature before the stop solution was applied. Optical density was read at 450 nm. Luminex Metabolic panel serum analyses of insulin, and c-peptide were performed using a Mouse Metabolic Hormone Magnetic Bead Panel (Millipore, MMHMAG-44K). Experimental protocols for the sample collection, storage, and preparation of reagents for immunoassay and immunoassay procedure, followed the specific instructions of the MMHMAG-44K mouse panel supplier. Results were read on a Luminex 200 analyzer with software (Xponent/Analyst version 4.2) used for data analysis. Insulin profile was also analyzed in serum collected from mice in the fed state.

## Metal ion quantification

Assays were performed by the Louisiana Animal Disease Diagnostics Laboratory with an Agilent Technologies 240 FS Atomic Absorption Spectrometer, in flame mode. Serum samples are quantitatively diluted in deionized water and subsequently analyzed. For the serum samples, a Seronorm Trace Elements Serum (L-2) is used as reference. First tissue samples are weighed and digested in nitric acid overnight at 85 °C. The following day, the samples are cooled down to room temperature and quantitatively transferred to polystyrene tubes with deionized water, and subsequently analyzed. For all tissue samples, a bovine liver standard reference material (SRM 1577 c) from the National Institute of Standards and Technology was used as a reference.

## Liver histology and histopathologic analysis

Explanted liver samples were fixed in 10% phosphate-buffered formalin acetate at 4 °C overnight, thoroughly rinsed in phosphate-buffered saline, and transferred to 70% ethanol. Histology was performed by HistoWiz Inc and Histoserv Inc using standard operating procedures and a fully automated workflow. Samples were embedded in paraffin wax and sectioned (5 μm). Prior to staining, slides were deparaffinized in xylene and hydrated with graded alcohols and finally water. Slides were then stained with either hematoxylin & eosin (H&E) or Picrosirius Red. Immunohistochemistry was performed on a Bond Rx autostainer (Leica Biosystems) with heat-induced epitope retrieval. Slides were incubated with primary antibodies F4/80 (Thermo, #14-4801-82), α-smooth muscle actin (Abcam, #ab5694), and Bond Polymer Refine Detection (Leica Biosystems) was used per manufacturer's protocol. Following staining, slides were dehydrated and coverslipped using a TissueTek-Prisma and Coverslipper (Sakura). Whole slide scanning (40 x) was performed on an Aperio AT2 (Leica Biosystems). For lipid staining, samples were frozen in O.C.T. compound (Tissue-Tek, #4583) and 5 μm thick sections were used. Slides were stained with Oil Red O (ORO) and Mayers hematoxylin and mounted with glycerin jelly. NAFLD scoring was performed by one external pathologist (provided by Histowiz) and one internal pathologist blinded to the samples, according to criteria described by *Liang et al., 2014*. Macrovesicular steatosis (H&E, ORO), hepatocyte hypertrophy (H&E), inflammation (H&E, F4/80), and fibrosis (PSR) were scored ranging from 0 to 3. NAFLD activity score is the sum of steatosis, hepatocyte hypertrophy, and inflammation scores. Steatosis-activity-fibrosis score is the sum of the NAFLD activity score and fibrosis score.

## Hepatic triglyceride assay

Lipids were extracted from liver samples using the Folch method (*Folch et al., 1957*) and solubilized as described earlier (*Carr et al., 1993*). The levels of triglyceride were measured using a Pointe triglyceride (GPO) reagent set (MedTest Dx, #T7532) and normalized to wet tissue weight.

## Glucose and insulin tolerance tests

An oral glucose tolerance test was administered upon an overnight fast (16 hr) with free access to water. Dextrose (Hospira Inc, NDC 0409-4902-34) was administered by oral gavage per 2 g/kg of body weight. Blood glucose was evaluated at defined time points (0, 15, 30, 60, and 120 min) using the AlphaTrak blood glucose monitoring system (Zoetis United States, Parsippany NJ) by sampling blood from the lateral tail vein. Insulin tolerance tests were performed after a 4 hr fast by administering 1.0 U/kg of body weight of Humulin R (Eli Lilly, #HI-213) by intra-peritoneal injection. Blood glucose was again evaluated at defined timepoints with the AlphaTrak blood glucose monitoring system by sampling blood from the lateral tail vein.

## HOMA-IR

Homeostatic model assessment of insulin resistance (HOMA-IR) indicates the level of insulin sensitivity by taking into account the relationship between glucose and insulin. HOMA-IR was calculated according to the formula: fasting insulin (microU/L) × fasting glucose (nmol/L)/22.5.

## Immunoblotting

For biochemical analysis, liver samples were harvested and immediately snap-frozen in liquid nitrogen. Protein was later extracted using RIPA buffer (Cell signaling technology, #9806) with Halt Protease & Phosphatase Inhibitor Cocktail (ThermoFisher Scientific, #78440). Protein concentration was determined using a Pierce TM BCA protein assay kit (Thermo Scientific, #23225). Five micrograms of protein of each sample were separated in NuPAGE 4–12% Bis-Tris protein gel (Invitrogen, #WG1403BOX), and transferred to nitrocellulose membrane using Trans-Blot Turbo Transfer System (BioRad). The membranes were blocked with 5% non-fat dry milk (BioRad, #9999) for 1 hr at room temperature before being incubated with primary antibody overnight at 4 °C. Antibodies were purchased from Cell Signaling Technology, phospho-AKT (Ser473, #4060), AKT (#9272), phospho-AMPKα (Thr172, #2535), AMPKα (#5831), phospho-LKB1 (Ser428, #3482), LKB1 (#3047), phospho-ACC (Ser79, #3661), ACC (#3676), FASN (#3189), HRP-linked anti-rabbit IgG (#7074), and HRP-linked anti-mouse IgG (#7076). Antibodies for G6PC (Invitrogen, #PA5-42541), SLC39A5, and β-actin (Sigma #5441) were used. For detection of SLC39A5 protein, liver samples were immunoprecipitated using Pierce Protein A/G Magnetic Beads and anti-SLC39A5 antibodies (Invitrogen, #88803, 42522), and eluted for western blot analysis. All membranes were washed before incubation with HRP-linked secondary antibody for 1 hr at room temperature. Blots were developed using SuperSignal West Femto Substrate (Thermo Fisher Scientific, #34095). Signals were captured using G:Box Mini 9 (Syngene). Densitometry analysis of immunoblots was performed using ImageJ.

## Gene expression analysis

Tissues were preserved in RNAlater solution immediately following harvest. Total RNA was purified using MagMAX–96 for Microarrays Total RNA Isolation Kit (Invitrogen, #AM1839) according to the manufacturer's specifications. Genomic DNA was removed using MagMAXTurboDNase Buffer and TURBO DNase from the MagMAX kit listed above. mRNA (Up to 2.5 ug) was reverse-transcribed into cDNA using SuperScript VILO Master Mix (Invitrogen, #11755500). cDNA was diluted to 0.5–5 ng/uL. 2.5–25 ng cDNA input was amplified with the SensiFAST Hi-ROX MasterMix (BIOLINE, #CSA-01113) using the ABI 7900HT Sequence Detection System (Applied Biosystems). The sequences of primers are as follows: *Slc39a5* (F 5'-CGAGCCTAGACCTCTTCCA- 3', R 5'-GGGAGCCATTCAGACA ATCC-3'), *Mt1* (F 5'-CAAGTGCACCTCCTGCAAGAAG-3', R 5'-CACAGCCCTGGGCACATTT-3'), *Mt2* (F 5'-GACCCCAACTGCTCCTGTG-3', R 5-'CTTGCAGGAAGTACATTTGCATTG-3'), *G6pc* (F 5'-GGTC GTGGCTGGAGTCTTG-3', R 5'-CCGGAGGCTGGCATTGTAG-3') and *Fasn* (Thermo Fisher Scientific #Mm00662319_m1).

## Human hepatocyte culture

Human Plateable Hepatocytes were purchased from Invitrogen (#HMCPP5, Lot. HPP1881027 and HPP1878738) and used according to manufacturer protocol. These Hepatocytes are a pooled population of primary hepatocytes produced by combining cells from five individual donors. All reagents and materials were purchased from Invitrogen. Briefly, cryopreserved hepatocytes were thawed in a hepatocyte thawing medium (#CM7500). Hepatocytes were centrifuged and resuspended in a plating medium, Williams' Medium E (#A1217601) with hepatocyte plating supplement (#CM3000). Hepatocytes were directly plated in collagen I coated 24-well plate (#A1142802). After 6 hr of incubation, media were replaced with incubation medium, Williams' Medium E with hepatocyte maintenance supplement (#CM4000). Next day, hepatocytes were treated with $ZnCl_2$ or $MgCl_2$ at the concentrations of 100, 200, and 400 uM in an incubation medium for 4 hr. Magnesium was used as a negative control, given that zinc and magnesium have opposite roles in the activation of protein tyrosine phosphatase 1B (*Bellomo et al., 2018*). In addition, Okadaic acid (OA) and Metformin (Met) were used as positive controls. OA is an inhibitor of the serine/threonine protein phosphatases (PP2A and PP1), resulting in an elevation of p.AMPKα Thr172 and p.AKT Ser473 levels in hepatocytes (*Samari et al., 2005*; *Galbo et al., 2011*). Metformin is an

antidiabetic drug that induces phosphorylation of AMPK in the liver (*Howell et al., 2017*). Protein lysates were collected using RIPA buffer with Halt Protease & Phosphatase Inhibitor and subjected to immunoblotting. Cell Viability assay was performed using CellTiter 96 AQueous One Solution Cell Proliferation Assay (MTS) assay (Promega, #G3580) per the manufacturer's protocol. HepG2 was purchased from ATCC (#HB-8065; The cell line was authenticated by STR profiling and tested negative for mycoplasma contamination, according to ATCC technical document) and cultured according to ATCC's protocol.

## Generation of Slc39a5 plasmids

Mouse *Slc39a5* ORF sequence was cloned into pIRES2 DsRed-Express2 vector (Clontech, #PT4079-5). Constructs of SLC39A5 variants were generated using the site-directed mutagenesis method, with oligos for Y47X (F 5'CCCATTCTCGCCCTACAGGCCAAACAGCTG-3', R 5-CAGCTGTTTGGCCTGT AGG GCGAGAATGGG-3'), R311X (F 5'-GGCCTGAGCCCTCAGTGCCGCAAAAGC-3', R 5'-GCTT TTGCGGCACTGAGGGCTCAGGCC-3'), R322X (F 5'-GTTTCGAGATTCCTTCAT TTTCGCCTGCAG-CATCT-3', R 5'-AGATGCTGCAGGCGAAAATGAAGGAATCTCGAA AC-3') and M304T (F 5'-AAAG CCCCAGCGTGTTCTCCAGCACAAAGAGCA-3', R 5'-TGCTCTTTGTGCTGGAGAACACGCTGGG GCTTT-3'). Mutagenesis was confirmed by Sanger sequencing.

## Membrane localization of SLC39A5 using flow cytometry

HEK293 cells (ATCC, #CRL-1573; The cell line was authenticated by STR profiling and tested negative for mycoplasma contamination, according to ATCC technical document) were plated on a 10 cm dish at a density of 25,000 cells/cm$^2$ and incubated at 37 °C at 5% $CO_2$ overnight in high glucose DMEM with 10% FBS and 1% Penn Strep (Gibco, #11965092). Cells were transfected the next day with 10 ug plasmid DNA using Xtremegene HP (Roche, #06366244001) in Opti-MEM (Gibco, #31985–070). Transfection complexes were incubated at room temperature for 20 min and added dropwise to the cells. Cells were incubated at 37 °C. A single media change was performed after 48 hr using Gibco DMEM high glucose and no other supplements. Following 24 hr of incubation at 37 °C cells were washed with DPBS and dissociated with Cell dissociation Buffer (Gibco, #13151–014). Cells were resuspended in 0.5% BSA (Sigma, #A7030) in DPBS and centrifuged at 200 g for 5 min. BD Biosciences CytoFix Fixation buffer (BD, #554655) was added to each sample and incubated at 4 C for 20 min. Cells were washed and stained with either anti-human SLC39A5 (Sigma, #SAB1408465) or isotype control followed by secondary Alexa 488 (Thermo Fisher, #A28175). Cells were washed and re-suspended in 0.5% BSA in DPBS. All samples were analyzed on a BD FACS CantoII.

## *MRE*-luc assay

HEK293 cells were plated in a 96-well plate at a density of 22,000 cells/well. Cells were transfected with MRE-luc (Promega pGL4.40), hRluc (Promega pGL4.75), and *Slc39a5* constructs with Xtreme-Gene HP (Roche) transfection reagent. MRE-binding transcription factor 1 (MTF1) is responsible for the expression of metallothioneins (MTs) in response to zinc. After 24 hr, the medium was changed with 0.5% charcoal stripped FBS in DMEM + Glutamax (Life Technologies, #10569010). Twenty-four hours later, the cultures were exposed to $Zn^{2+}$ for 6 hr. Luciferase activity was measured using a Dual-Glo luciferase assay system (Promega, #E2920). Results were expressed as the relation of firefly luciferase activity to Renilla luciferase activity.

## Live staining & immunofluorescence

HEK293 cells were plated on Ibidi chamber slides (Ibidi, #80427) coated with 100 µg/mL of poly-L-lysine. Live cell staining was performed by diluting the primary antibody (Polyclonal anti-SLC39A5, Sigma SAB1408465, 1:100) in a cold culture medium and incubated with the cells for 2 hr at 4 °C. Slides were washed with cold PBS 3×10 min at 4 °C with gentle shaking. Cells were fixed with 2% paraformaldehyde at room temperature for 5 min and washed with PBS. Wheat germ agglutinin in HBSS was added to cells for 10 min at room temperature and then blocked with 5% goat serum. Secondary antibody was applied for 45 min at room temperature. Cells were washed and mounted with ProLong diamond antifade solution and imaged using Zeiss AxioObserver LSM880.

## Hepatic β-hydroxybutyrate assay

β-OHB (Sigma, #MAK041) was measured in mouse liver tissue by colorimetric assays per manufacturer's guidelines. Briefly, liver samples (10 mg/sample) were homogenized in 4 vol of cold β-OHB assay buffer. Samples were centrifuged at 13,000 g for 10 min at 4 °C and supernatant was used for analyses. Concentration is determined by a coupled enzyme reaction which results in a colorimetric (450 nm) product proportional to the β-OHB present.

## Protein phosphatase assay

Cultured hepatocyte lysates and tissue homogenates were prepared by Lysis Reagent (Invitrogen, #M10510) with protease inhibitor cocktail (Sigma, #P8340). Protein lysates were centrifuged at 4 °C for 15 min and protein concentration was determined using Pierce TM BCA protein assay kit. Phosphatase activity was measured using RediPlate 96 EnzChek Tyrosine Phosphatase and Serine/Threonine Phosphatase Assay Kit (Invitrogen, #R22067, R33700) per manufacturer's instructions. Samples were incubated in RediPlate for 60 mins before reading by fluorescence microplate with excitation/emission at 358/455 nm (Spectramax M4, Molecular Devices).

## Superoxide dismutase (SOD) assay

SOD activity was measured in mouse liver tissues using a SOD activity kit (ENZO, #ADI-900–157). Briefly, liver samples (30 mg/sample) were homogenized in 10 vol of cell extraction buffer. Samples were centrifuged at 10,000 g for 10 min at 4 °C and supernatant was used for analysis. Protein concentration was determined by Pierce TM BCA protein assay kit (Thermo Scientific, #23225). Samples were diluted to 50 ug/25 ul for assay, according to protocol.

## Statistical analysis

Results are shown as box plots with individual values. Statistical analysis was performed using GraphPad Prism 8 software. Analysis for $Slc39a5^{-/-}$; $Lepr^{-/-}$ mice and corresponding control mice was performed using one-way ANOVA, followed by post hoc Tukey's tests. Analysis for HFFD and NASH mouse studies was performed using two-way ANOVA, followed by post hoc Tukey's tests. Statistical significance reported when $p<0.05$. Sample sizes, statistical tests, and significance are described in each figure legend.

# Acknowledgements

We wish to thank individuals that consented to be part of the Regeneron Genetics Center-Geisinger Health System DiscovEHR study. We would also like to acknowledge Kristy Neiman and William Poueymirou for their assistance with mouse colony management, and Suganthi Balasubramanian for insightful comments and input at the inception of the project, and Sergio Fazio for a critical reading of our manuscript. This work was supported by Regeneron Pharmaceuticals.

# Additional information

#### Competing interests

Shek Man Chim, Kristen Howell, John Dronzek, Weizhen Wu, Manuel AR Ferreira, Bin Ye, Alexander Li, Susannah Brydges, Vinayagam Arunachalam, Anthony Marcketta, Adam E Locke, Jonas Bovijn, Niek Verweij, Tanima De, Lyndon Mitnaul, Michelle LeBlanc, Regeneron Genetics Center, Alan Shuldiner, Aris N Economides, Harikiran Nistala: full-time employee of the Regeneron Genetics Center or Regeneron Pharmaceuticals Inc and hold stock options/restricted stock as part of compensation. Luca Lotta: full-time employee of the Regeneron Genetics Center or Regeneron Pharmaceuticals Inc and hold stock options/restricted stock as part of compensation.full-time employee of the Regeneron Genetics Center or Regeneron Pharmaceuticals Inc and hold stock options/restricted stock as part of compensation. DiscovEHR collaboration: Regeneron Genetics Center: The other authors declare that no competing interests exist.

## Funding

| Funder | Grant reference number | Author |
|---|---|---|
| Regeneron Pharmaceuticals | | Shek Man Chim<br>Kristen Howell<br>John Dronzek<br>Weizhen Wu<br>Cristopher Van Hout<br>Manuel AR Ferreira<br>Bin Ye<br>Alexander Li<br>Susannah Brydges<br>Vinayagam Arunachalam<br>Anthony Marcketta<br>Adam E Locke<br>Jonas Bovijn<br>Niek Verweij<br>Tanima De<br>Luca Lotta<br>Lyndon Mitnaul<br>Michelle LeBlanc<br>David J Carey<br>Olle Melander<br>Alan Shuldiner<br>Katia Karalis<br>Aris N Economides<br>Harikiran Nistala |

The funders had no role in study design, data collection and interpretation, or the decision to submit the work for publication.

## Author contributions

Shek Man Chim, Harikiran Nistala, Conceptualization, Data curation, Supervision, Investigation, Writing – original draft, Writing – review and editing; Kristen Howell, John Dronzek, Weizhen Wu, Data curation, Investigation; Cristopher Van Hout, Manuel AR Ferreira, Bin Ye, Alexander Li, Susannah Brydges, Vinayagam Arunachalam, Anthony Marcketta, Adam E Locke, Jonas Bovijn, Niek Verweij, Tanima De, Luca Lotta, Lyndon Mitnaul, David J Carey, Olle Melander, Alan Shuldiner, Investigation; Michelle LeBlanc, Project administration; Regeneron Genetics Center, Resources; Katia Karalis, Supervision, Writing – original draft, Writing – review and editing; Aris N Economides, Conceptualization, Supervision, Writing – original draft, Writing – review and editing; DiscovEHR collaboration, Data curation, Resources; Regeneron Genetics Center, Conceptualization, Formal analysis, Investigation, Methodology, Project administration, Supervision, Validation, Visualization, Writing – original draft, Writing – review and editing

## Author ORCIDs

Shek Man Chim (ID) https://orcid.org/0000-0002-5116-8394
Adam E Locke (ID) https://orcid.org/0000-0001-6227-198X
Aris N Economides (ID) https://orcid.org/0000-0002-6508-8942
Harikiran Nistala (ID) http://orcid.org/0000-0003-4928-7527

## Ethics

All experimental protocols in mice including anesthesia and tissue sampling procedures performed in this study, were approved by Regeneron Pharmaceuticals Inc Institutional Animal Care and Use Committee (IACUC) under protocol number 430 and the US Animal Welfare Act.

Reviewer #1 (Public Review): https://doi.org/10.7554/eLife.90419.2.sa1
Reviewer #2 (Public Review): https://doi.org/10.7554/eLife.90419.2.sa2
Author response https://doi.org/10.7554/eLife.90419.2.sa3

# Additional files

## Supplementary files

• Supplementary file 1. Serum zinc and insulin profile assessment in the serum call back study. Serum

zinc levels in SLC39A5 heterozygous loss of function carriers are elevated by 12% as compared to age, sex, and BMI-matched reference controls. Analyses of insulin production (insulin/c-peptide ratio), insulin clearance (proinsulin/insulin), and blood glucose in these samples demonstrated no differences based on genotype. Data is represented in a graphical format in *Figure 1*.

• Supplementary file 2. Tissue zinc content in humans and mouse. Human data adapted from *Jackson et al., 1982* *p<0.05; **p<0.01, ***p<0.001, not significant (n.s.), unpaired t-test. Values represent mean ± SD.

• Supplementary file 3. No differences in the serum chemistry profile of Slc39a5$^{+/+}$ and Slc39a5$^{-/-}$ mice. Serum chemistry analysis in adult mice (40 wk of age, both sexes) demonstrated no differences in pancreatic amylase, renal function parameters (blood urea nitrogen, creatinine, total protein, and uric acid), and electrolytes (chloride, potassium, and sodium) or liver enzymes (alanine aminotransferase; ALT and aspartate aminotransferase; AST).

• Supplementary file 4. Summary statistics for the congenital obesity model. Loss of *Slc39a5* improves glycemic traits and liver function in leptin-receptor (*Lepr*) deficient mice. Loss of *Slc39a5* does not change insulin production (proinsulin/insulin), or insulin clearance (insulin/c-peptide ratio).

• Supplementary file 5. Summary statistics for the diet-induced obesity model. Loss of *Slc39a5* improves glycemic traits and liver function in mice upon a high-fat high fructose diet (HFFD) dietary challenge. Moreover, loss of *Slc39a5* does not change insulin production (proinsulin/insulin), or insulin clearance (insulin/c-peptide ratio).

• Supplementary file 6. Summary statistics for the diet-induced non-alcoholic steatohepatitis (NASH) model. Loss of *Slc39a5* improves hepatic inflammation and fibrosis in both female and male mice challenged with diet-induced NASH.

• MDAR checklist

## Data availability

All data generated or analysed during this study are included in the manuscript and supporting files; source data files have been provided for Figures 5, Figure 2-figure supplement 1, Figure 3-figure supplement 1, Figure 3-figure supplement 2, Figure 3-figure supplement 3, Figure 3-figure supplement 4, Figure 5-figure supplement 1, Figure 5-figure supplement 3.

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
