## [Editor Report · eLife assessment]

This **fundamental** study substantially advances our understanding of the role of zinc in metabolism, specifically a newly established clinical link between mutations in the zinc transporter SLC39A5, elevated serum zinc levels, and a reduction in the risk of Type 2 Diabetes. The provided evidence is **solid** with state-of-the-art genetic analysis of large human cohorts followed by a comprehensive analysis of a mouse SLC39A5 knockout mutant, establishing that SLC39A5 plays a role in hepatic lipid handling through AMPK signaling, although the limited analysis of a pancreatic phenotype that has previously been described constitutes a weakness. This study will be of relevance to researchers interested in metabolism, fatty liver disease, and the biology of trace elements.

---

## [Referee Report · Reviewer #1 (Public Review)]

Summary:

In their manuscript, Chim et al. identify an association of rare loss-of-function (LOF) SLC39A5 variants with increased circulating zink levels and decreased T2D risk and complement these observations with a notably comprehensive analysis of metabolically challenged (genetically or diet-induced) Slc39a5-/- mice that demonstrate enhanced hepatic zinc levels, improved liver function, reduced hyperglycemia, partial resistance to NASH induction, and likely involvement of AMPK and AKT signaling.

Strengths:

Overall, the work appears well designed, executed, clearly presented (although navigating the 16 supplementary figures and 6 supplementary tables can be a bit of a challenge), and supports the authors' main conclusions.

Weaknesses:

Nevertheless, two major concerns pertain to the characterization of LOF SLC39A5 variants as well as the seeming absence of a "pancreatic phenotype" in Slc39a5-/- mice that contrasts with earlier reports including impaired glucose tolerance and glucose-stimulated insulin secretion in mice lacking Slc39a5 specifically in beta cells; these concerns should be addressed experimentally and by more extensive discussion of previously published Slc39a5-/- mouse models, respectively.

---

## [Referee Report · Reviewer #2 (Public Review)]

Summary:

This study links rare human loss of function mutations in the zinc transporter family member SLC39A5 to increased circulating and hepatic concentrations of this trace element. Beneficial metabolic changes were observed in a corresponding convincing mouse model relevant to the development of NASH.

Strengths:

Authors combine human exome sequencing data, meta-analysis of four large European cohorts, and a patient recall approach to link the rare loss of function variants of SLC39A5 to the phenotype and protection from T2DM.

Using a SLC39A5-null mouse model challenged either by cross-breeding with Lepr-/- mice or diet-induced obesity they unravel the metabolic impact of elevated circulating and hepatic zinc concentration with respect to T2DM, glucose homeostasis, hepatic steatosis, and NASH development. Some mechanistic aspects and a remarkable sex difference in the outcome are identified from mouse ex vivo readouts and supported by in vitro hepatocyte cellular studies. Authors present evidence that increased hepatic zinc concentrations inhibit zinc-regulated phosphatases resulting in activation of AMPK and AKT signalling with consequences for lipid and glucose metabolism and insulin sensitivity.

Weaknesses:

The reasons for the observed sex differences in the metabolic consequences of SLC39A5 inactivation in the mouse models remain unclear. While heterozygous rare SLC39A5 variants show distinct phenotypes only SLC39A5-null mice and no heterozygous mice are studied. The role of SLC39A5 in pancreatic islets and on insulin secretion remains unclear because authors do not address data published recently that claim a relevant role of SLC39A5 in b-cell function and glucose tolerance.

---

## [Author Response]

Pancreatic phenotype reported by Wang et al., 2019 (PMID 30324491)

The reported human knockout of SLC39A5 (homozygous for R311* allele) suggests that SLC39A5 is dispensable for embryonic development with no adverse effect on postnatal pancreatic development or function (Saleheen D, 2017). Indicative of conserved expression and function, Slc39a5 is non-essential in mice, with homozygous or heterozygous deletion of Slc39a5 resulting in elevated serum zinc (Fig. 2) and no resulting impairment in pancreatic development or function (Fig. S3, S4E-F, S5E-F, S6E-F, S7E-F, S8A-H).

The observed antihyperglycemic effects in the Slc39a5 LOF animals were not driven by changes in insulin production and/or clearance (Fig. S3, S4E-F, S5E-F, S6E-F, S7E-F, S8A-H). Our observations related to pancreatic function (both exocrine and endocrine; Fig. 3 and Suppl. Table 3-5) in the Slc39a5 LOF mice are in agreement with reported metabolic phenotyping by the International Mouse Phenotyping Consortium (https://www.mousephenotype.org/data/genes/MGI:1919336). Intriguingly, Wang et al. reported impaired insulin secretion in mice with Ins2-cre mediated deletion of Slc39a5 in β-cell cells (Wang X, 2019). These findings are difficult to interpret in light of single cell RNA-seq analyses of mouse pancreas demonstrating absence of Slc39a5 expression in Ins2+ pancreatic β-cells (The Tabula Muris Consortium, 2018 and 2020). Consistently, SLC39A5 expression in human pancreas is largely restricted to pancreatic acinar and ductal cells (Baron M, 2016; Muraro MJ, 2016; Xin Y, 2016).

Taken together, these observations suggest that the protective metabolic changes are presumably extra-pancreatic in both mouse and human.

Sex Differences:

Slc39a5 LOF activates hepatic AMPK signaling in both sexes, hepatic AKT signaling is elevated in females, suggesting that the observed glucose lowering effects in the Slc39a5 LOF male mice is possibly driven by improvements in extra hepatic glucose metabolism in males or that the magnitude of zinc mediated protein phosphatase inhibition is insufficient to influence the hepatic PI3K/AKT signaling in males. Whether the promotion of hepatic AMPK and AKT signaling occurs solely as a result of zinc mediated inhibition of protein phosphatases or a result of concurrent convergent mechanisms potentially influenced by sex hormones remains to be resolved in future investigations.

Overall, integrated analyses of the metabolic phenotyping in our models (both diet-induced and congenital obesity) are consistent with the well documented sex-dependent susceptibility to obesity-related metabolic alterations such as insulin resistance, hepatic steatosis and dyslipidemia (Goodpaster BH, 2003; Priego T, 2008; Medrikova D, 2012; Bertolotti M, 2014; Frias JO 2001; Krotkiewski M, 1983; Lebeck J, 2016).